# Two-years antibody responses following SARS-CoV-2 infection in humans: A study protocol

Eggi Arguni[1,2], Fatwa Sari Tetra Dewi[2,3], Jajah Fachiroh[2,4], Dewi Kartikawati Paramita[4], Septi Kurnia Lestari[2], Bayu Satria Wiratama[5], Annisa Ryan Susilaningrum[2], Bara Kharisma[2], Yogi Hasna Meisyarah[2], Merlinda Permata Sari[2], Zakiya Ammalia Farahdilla[2], Siswanto Siswanto[6], Muhammad Farhan Sjaugi[7], Teguh Haryo Sasongko[8], Lutfan Lazuardi[9]*

1 Department of Child Health, Faculty of Medicine, Public Health and Nursing, Universitas Gadjah Mada, Yogyakarta, Indonesia, 2 Health and Demographic Surveillance System Sleman, Faculty of Medicine, Public Health and Nursing, Universitas Gadjah Mada, Yogyakarta, Indonesia, 3 Department of Health Behavior, Environment, and Social Medicine, Faculty of Medicine, Public Health and Nursing, Universitas Gadjah Mada, Yogyakarta, Indonesia, 4 Department of Histology and Cell Biology, Faculty of Medicine, Public Health and Nursing, Universitas Gadjah Mada, Yogyakarta, Indonesia, 5 Department Biostatistics, Epidemiology and Population Health, Faculty of Medicine, Public Health and Nursing, Universitas Gadjah Mada, Yogyakarta, Indonesia, 6 Universitas Gadjah Mada Academic Hospital, Universitas Gadjah Mada, Yogyakarta, Indonesia, 7 Perdana University Graduate School of Medicine and Perdana University Center for Research Excellence, Kuala Lumpur, Malaysia, 8 Department of Physiology, School of Medicine and Institute for Research, Development, and Innovation, International Medical University Kuala Lumpur, Malaysia, 9 Department of Health Policy and Management, Faculty of Medicine, Public Health and Nursing, Universitas Gadjah Mada, Yogyakarta, Indonesia

* lutfan.lazuardi@ugm.ac.id

**Data Availability Statement:** No datasets were generated or analysed during the current study. All relevant data from this study will be made available upon study completion.

## Abstract

The long-term antibody response to the novel SARS-CoV-2 in infected patients and their residential neighborhood remains unknown in Indonesia. This information will provide insights into the antibody kinetics over a relatively long period as well as transmission risk factors in the community. We aim to prospectively observe and determine the kinetics of the anti-SARS-CoV-2 antibody for 2 years after infection in relation to disease severity and to determine the risk and protective factors of SARS CoV-2 infections in the community. A cohort of RT-PCR confirmed SARS-CoV-2 patients (case) will be prospectively followed for 2 years and will be compared to a control population. The control group comprises SARS-CoV-2 non-infected people who live within a one-kilometer radius from the corresponding case (location matching). This study will recruit at least 165 patients and 495 controls. Demographics, community variables, behavioral characteristics, and relevant clinical data will be collected. Serum samples taken at various time points will be tested for IgM anti-Spike protein of SARS-CoV-2 and IgG anti-Spike RBD of SARS-CoV-2 by using Chemiluminescent Microparticle Immunoassay (CMIA) method. The Kaplan-Meier method will be used to calculate cumulative seroconversion rates, and their association with disease severity will be estimated by logistic regression. The risk and protective factors associated with the SARS-CoV-2 infection will be determined using conditional

**Funding:** The study has received financial grant support from the Ministry of Finance of the Republic of Indonesia LPDP (RISPRO/KI/B1/TKL/5/15867/2/2020). EA received the award. The funders had and will not have a role in study design, data collection and analysis, decision to publish, or preparation of the manuscript.

**Competing interests:** The authors have declared that no competing interests exist.

(matched) logistic regression and presented as an odds ratio and 95% confidence interval.

## Introduction

It has been over two years since the currently ongoing Coronavirus Disease 2019 (COVID19) pandemic was announced as a public health emergency of international concern. COVID-19 has inflicted tremendous mortality and morbidity in a relatively short time [1–3]. The duration of a protective immune response after a Severe Acute Respiratory Syndrome Coronavirus 2 (SARS-CoV-2) infection has not yet been fully studied. Still, the dynamics of the humoral immune response during an acute phase of SARS-CoV-2 infection are well-understood [4, 5].

The seroconversion takes place 6–14 days after the diagnosis of SARS-CoV-2 infection [6–9]. A significant increase in virus-specific antibody levels is observed on days 16–35 after the onset of symptoms [7, 10–13]. Some studies have reported that antibody levels peak within the first few months, then wane and remain stable for several months, indicating that the immunity may last longer [7, 11, 13–19]. Other studies suggest that the levels of SARS-CoV-2-specific IgG are durable and decline after 6–8 months [20, 21]. The latest study by Marcotte et al. reported that anti-SARS-CoV-2 RBD and S IgG increased at 15–28 days after symptoms onset in almost all of the patients, then gradually dropped by less than 4-fold from the antibody response until 6 months, but thereafter, persist in the majority of patients up to 15 months [22]. Tan et al. reported that IgM was detected on day 7 (peaked on day 28), and IgG was detected on day 10 (peaked on day 49) [23]. A previous study by Zhao et al. determined that seroconversion took place at later median times [24].

Severity levels of cases, e.g., asymptomatic, mild, or severe, were found to produce identifiable antibodies against MERS-CoV, SARS-CoV and SARS-CoV-2 [25]. Although differences in disease severity do not lead to significant kinetic changes of the antibodies, severe SARS-CoV cases with sequelae showed lower neutralizing antibodies [26]. Seroconversion rate, as well as peak antibody levels, increased according to disease severity in MERS-CoV infection, [27] and antibodies response seemed to be less likely to evoke in milder infections [27, 28]. Tan et al. showed that SARS-CoV-2 infection might indicate a probable opposing pattern to MERS-CoV, for which in both severe and non-severe patients, IgM emerged at the same time, while IgG emerged earlier in severe patients [23]. Marcotte et al. reported that the half-lives of anti-RBD and anti-S IgG antibody response were shorter in patients with mild/moderate than severe/critical disease [22].

Furthermore, in the general population, understanding risk factors for COVID-19 is needed to plan an appropriate disease prevention model. A surveillance system is needed to accommodate various challenges in data collection involving hundreds of respondents. Seroconversion and duration of immune response among patients with COVID-19 in Indonesia, particularly in a cohort setting, have never been reported.

This study protocol describes plans to observe and determine the kinetics of the antibody response up to two years following SARS-CoV-2 infection and identify the protective and risk factors of infection in the community. The ability to predict the likelihood of SARS-CoV-2 infection has the potential to inform decision-making at the individual, stakeholder, and government levels. Furthermore, evidence-based risk stratification tools can provide a platform for policymakers to prioritize population segments in national vaccination programs.

## Methods

### Aims of the study

The primary objective of this study is to prospectively observe and determine the kinetics of anti-SARS-CoV-2 antibodies for 2 years after infection in relation to disease severity. The secondary objective is to identify the risk and protective factors of SARS CoV-2 infections through a comparison of SARS-CoV-2 confirmed patients (case group) and their respective neighborhood individuals (control group). Further, we aim to develop a cohort system in managing responses to the pandemic, which will combine hospital-based and population-based longitudinal data in the Sleman District, Yogyakarta Province, Indonesia. Such a cohort system will contribute to the development of a fast, accurate, and secure data collection and management application.

### Study design

This study comprises two sub-studies, i.e., a case-control study and a prospective cohort study. We will use the case-control design to determine the risk and protective factors of SARS-CoV-2 infections. Cases will be recruited from the Academic Hospital of the Universitas Gadjah Mada (UGM) in Yogyakarta. For each case, 3 control subjects, who are uninfected, will be recruited from an area within a 1-km radius from the case's residence. We hypothesized that individual behavior, traveling, and societal structures might affect the risk of infection. Then, we will implement the prospective cohort design to observe the kinetics of antibody anti-SARS-CoV-2 in cases for two years following infection. We will use Chemiluminescent Microparticle Immunoassay (CMIA) method to quantify the peripheral blood IgM/IgG anti-Spike RBD titer of SARS-CoV-2. The case-cohort will provide parallel data, enabling us to correlate the seroconversion and dynamics of antibody responses with clinical data, demographics, and comorbidities. We hypothesized that the IgM/IgG titer to SARS-CoV2 will correlate with disease severity.

### Study setting

The study is taking place in Sleman, a district within the Yogyakarta province, Indonesia. The district spans an area of 574.82 $km^2$, which is about 18% of the province and is inhabited by 1,219,640 residents (December 2020). Geographically, Sleman is located between 110° 33′ 00″ and 110° 13′ 00″ East Longitude, 7° 34′ 51″ and 7° 47′ 30″ South Latitude (Fig 1) [29]. As of 31 August 2021, the Sleman District COVID-19 Task Force reported cumulative cases of 52,766, of which 6,131 were active, with 2,308 deaths [30].

### Sample size

Fleiss's formula was used to calculate the sample sizes of cases and controls with a ratio of 1:3 with 90% power and 95% confidence interval (CI) to detect an odds ratio of 2 [31]. The resulted sample size was 118 cases. The sample size was further adjusted to account for the expected rate of loss to follow-up and missing data of 40%. Thus, the final sample size for cases is 165, and 495 healthy individuals for control will be recruited from the Sleman population.

### Study participants

Patients will be identified prospectively at the time of confirmation of the diagnosis in isolation wards at the UGM Academic Hospital. The inclusion criteria are patients older than 18 years old who are confirmed COVID-19, defined as positive SARS-CoV-2 reverse transcriptase-

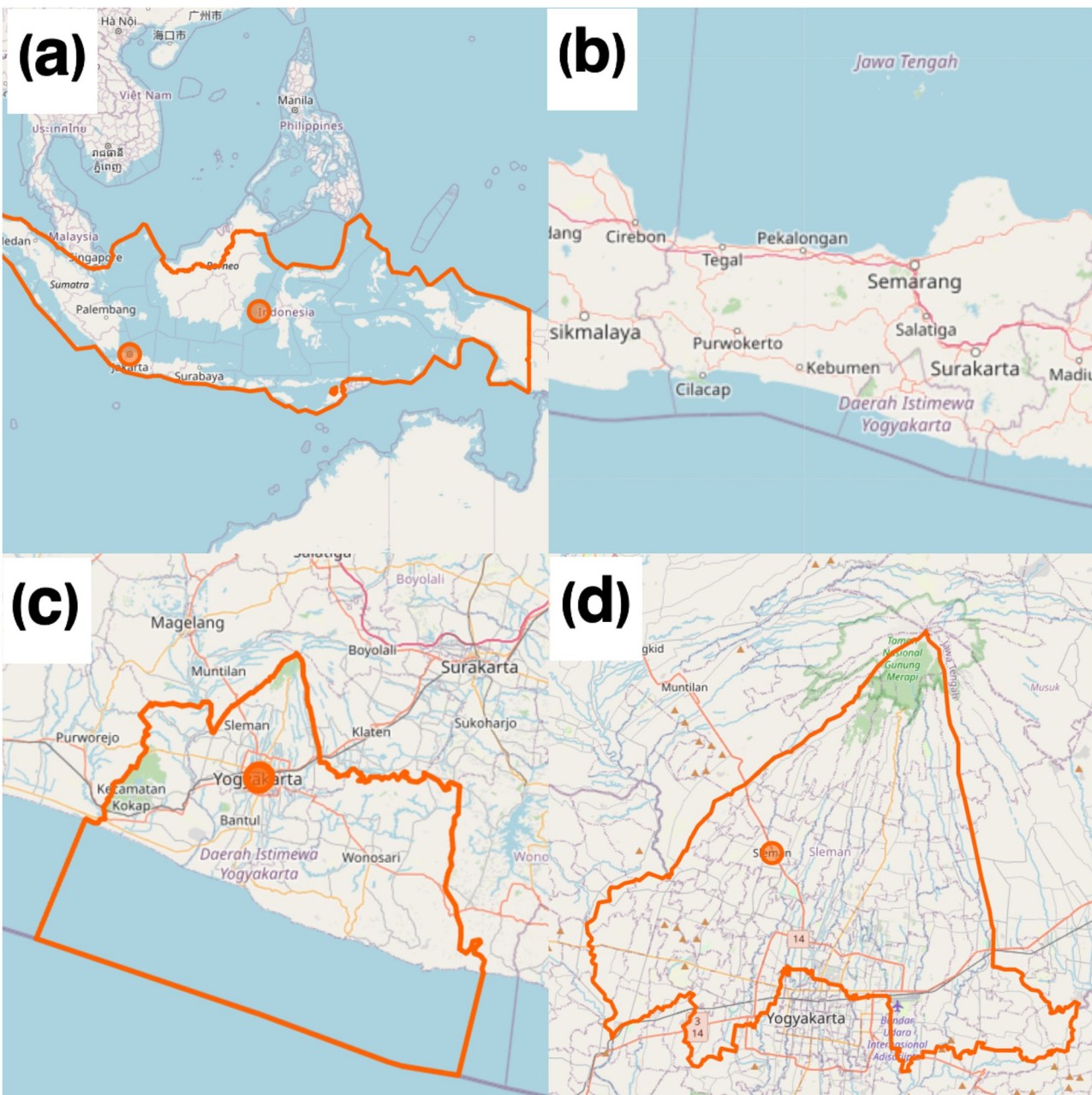

**Fig 1. Location of the research.** (a) the country of Indonesia (b) the island of Java (c) Yogyakarta province, and (d) Sleman regency © OpenStreetMap Contributor. This map is licensed under the Creative Commons Attribution-Share Alike 2.0 License (CC BY-SA). For further information, please visit http://www.openstreetmap.org/copyright.

polymerase chain reaction (RT-PCR) results from any respiratory sample, and reside in Sleman District at the time of diagnosis. There are no exclusion criteria for the case group.

The controls comprise individuals who live within a 1-km radius of the case's residence. The inclusion criteria are healthy adults (aged >18 years old) with no prior or current SARS-CoV-2 infection upon screening (using GeNose®) at the time of interview. The exclusion criteria are any eligible person who shows a flu-like syndrome on the screening day.

Control group sampling procedures will be done as follows: the case's residence will be located using Google Earth. The longitude and latitude of the location point will be documented and then imported into Google Earth. Subsequently, a radius of 1 kilometer will be made from the point before generating aerial imagery referenced to 100-meter grids within the radius. Each grid cell will be assigned a number consecutively according to row, with the smallest number, one, assigned to the top left corner cell and the largest number assigned to the bottom right corner cell. Five grid cells will be randomly selected (among which 3 will stand as a backup if the land is vacant or the number of houses is too few). Houses in the selected grids will be numbered. Twenty households will be selected by random sampling. The field staff will visit the selected houses starting from the first selected house. The Kish grid method will be used to select an individual in each house as a potential control subject [32, 33].

## Data collection

Demographic data (basic demographic data, mobility, travel history, number and composition of households, and house size), community variables (environmental density, local transmission in the neighborhood, contact history), behavior (adherence to health protocols, tobacco use, transportation, and sleep patterns), comorbidity (diabetes, cancer, stroke, hypertension, obesity, heart disease, chronic obstructive pulmonary disease, chronic liver disease, chronic kidney disease, and autoimmune disease), pregnancy and COVID-19 vaccination status, and anthropometric parameters will be collected from medical records (case group) and direct interviews (control group). The data collection process in the community will be a collaborative effort involving local stakeholders and volunteer cadre groups.

The Hospital's laboratory information of the case group (SARS-CoV2 RT-PCR of nasopharyngeal/oropharyngeal swab, complete blood count, electrolyte, blood glucose, ferritin, CRP, procalcitonin, liver transaminases, serum, BUN, serum creatinine, chest X-ray and/or computerized tomography (CT)-scan), clinical management (drugs, intravenous fluid, supplements, oxygenation, ventilator support, and other intensive care support), and comorbidity (diabetes, cancer, stroke, hypertension, obesity, heart disease, chronic obstructive pulmonary diseases, pregnancy) will be retrieved from hospital records. The severe group will be categorized as patients who are dyspnea, hypoxia (SpO$^2$ <93% in room air), require ICU level care and other organ support (such as inotropes and renal replacement therapy), or die after enrolment during hospitalization. Non-severe groups will be categorized as patients who do not meet the criteria for severe disease. Data of any resolution/persistence of symptoms, travel history, compliance with health protocols, any possible COVID-19 re-infection, and COVID-19 vaccination status will be recorded in each subsequent monitoring time point.

## Blood samples collection and storage

A 3 mL venous blood sample will be collected from all consenting participants. Serial peripheral blood samples will be collected on the day of diagnosis when discharged from the hospital and at weeks 2, 4, 6, 8, 10, 12, 16, 20, 24, 36, 48, 72, and 96 after discharge. For monitoring samples, the blood will be collected at the outpatient clinic or a nearby community health center by making an appointment beforehand to meet with the research nurse/research officer. Once collected in an EDTA-containing vacutainer, blood samples will be transferred to the project laboratory (Biobank Unit at the Faculty of Medicine, Public Health and Nursing UGM, Yogyakarta, Indonesia). Plasma will be collected by centrifugation at 4,000 rpm for 10 minutes [34]. All specimens will be processed and stored on the same day as the sample receipt. Plasma will be stored at -80 ˚C until testing.

## CMIA for IgM and IgG anti-SARS-CoV2

Examination of IgM anti Spike protein of SARS-CoV2 and IgG against Spike RBD of SARS-CoV-2 will be performed using Chemiluminescent Microparticle Immunoassay (CMIA) from Abbot (Cat. 6R87 for IgM and 6S60 for IgG). Two types of immunoassays are used in this study. Both are automated, two-step immunoassays. The SARS-CoV-2 IgM assay is used for the qualitative detection of IgM antibodies SARS-CoV2, and the SARS-CoV-2 IgG II Quant assay is used for the qualitative and quantitative determination of IgG antibodies against SARS-CoV-2. Serum or plasma (separated by dipotassium EDTA or tripotassium EDTA or lithium heparin or sodium heparin or sodium citrate) can be used in this assay. Both of the assays have an identical principle in the procedure. The sample, SARS-CoV2 antigen coated paramagnetic particles, and assay diluent are mixed and incubated. The IgM and IgG antibodies against SARS-CoV2 present in the sample will bind to the SARS-CoV2 antigen-coated microparticle. The detection of IgM and IgG are processed separately. After the washing step, anti-human IgM or IgG acridinium-labeled conjugate is added to the mixture and incubated. Following a wash cycle, Pre-Trigger and Trigger Solutions are added. The resulting chemiluminescent reaction is measured as a relative light unit (RLU) [35, 36].

## Data management plan

All respondents will have unique identification. Research nurse/field officers will enter data into eSynthesis (version 1.9.8—A survey tool. Sleman, Yogyakarta, Indonesia), an application installed on their mobile tablet. Supervisors and data managers will check the suitability of the data to ensure data quality. Maintaining data quality will include calibrating instruments regularly, and having a validation feature in the application. A red warning sign will appear if the input format does not match or if there are data that have not been filled in. The data will be cleaned several times by the data manager until ready for analysis. Data types consist of text, number, audio, and image data. The data consist of primary and secondary data. Primary data will be obtained by measuring directly and interviewing respondents in the field. Secondary data will be taken from the respondents' medical records. Data will be stored on institutional servers restricted to a particular research team. The study's data processing (collection, storage) is based on the UGM Academic Hospital and the Health and Demographic Surveillance System Sleman (HDSS Sleman) data protection concepts.

## Expected outcomes

We intend to measure the kinetics of antibody response to the receptor-binding domain (RBD) of the spike (S) protein of SARS-CoV-2 in a cohort of patients infected with SARS-CoV-2 (up to 2 years after detection of SARS-CoV-2 RNA). The association between antibody level and severity of illness will be measured across time. Risk and protective factors associated with SARS-CoV-2 infection will be measured at the time of case detection by comparing the proportion of factors between the case and control groups.

## The type of data and statistical analyses planned

Continuous variables with normal distribution will be reported as mean with standard deviation (SD), and non-normal distributed variables as median with interquartile range (IQR). The Kaplan-Meier method will be used to calculate cumulative seroconversion rates. The association between antibody level and disease severity will be estimated by logistic regression. The characteristics of the case and control groups will be described as mean and SD or frequency and percentage. The factors (risk and protective) associated with the COVID-19 infection will

be determined using conditional (matched) logistic regression and presented as OR and its 95% CI. This study will be reported in accordance with the Strengthening the Reporting Observational Studies in Epidemiology (STROBE) checklist [37]. Missing data and lost to follow-up patients will be reported. Missing data will be handled by using multiple imputation models [38].

### Ethical considerations and declarations

This study was reviewed and approved by the Medical and Health Research Ethics Committee (MHREC) Faculty of Medicine, Public Health, and Nursing UGM (KE/FK/0882/EC/2020). Written informed consent will be documented from all respondents prior to enrolment into the study. The authors are responsible for the content and writing of the paper.

### The status and timeline of the study

The study started enrolment in January 2021 and is recruiting a cohort of study participants. This multi-year research project will be conducted for three years with respective targets and indicators. The first and second years of the study focus on participant enrolment, monitoring, and laboratory analysis. Study results are planned to be disseminated in the third year of study.

## Discussion

Antibody-based detection can help determine prior SARS-CoV-2 infections in populations and may allow for vaccination monitoring in the future [39]. Examination of antibodies against SARS-CoV2 is important for several reasons: confirmation of present and past infection, evaluation of patients with characteristic COVID-19 symptoms but negative nucleic acid amplification tests (NAATs), seroepidemiological studies on COVID-19, assessment of the development of antibody-mediated protective immunity and investigation of the immune response and immunopathology of COVID-19.

This study is designed to observe the dynamics of antibody anti-SARS-CoV-2 for two years following SARS-CoV-2 infections. Potential key points relevant to the current pandemic include unknown seroconversion and seroreversion in hospitalized patients. Time to seroconversion may be related to the severity of the disease. A study from North America showed that in symptomatic COVID-19 patients, the median time to seroconversion of SARS-CoV-2 RBD (IgG and IgM) among hospitalized patients was 4 days earlier compared to non-hospitalized patients, proposing an association between antibody kinetics and disease severity [8]. SARS-CoV-2 RBD and S IgG can be detected at approximately the same time as IgM (usually within 2 weeks after symptom onset) [22]. SARS-CoV-2 RBD and S IgG titers peaked 14–30 days after infection. Even though they decreased over time, the IgG antibody response stabilized after 6 months and was still detected in the majority of patients at 6–15 months [22, 40–42].

IgG responses to the SARS-CoV-2 spike RBD showed minimal seroreversion, which was sustained for more than 3 months. Meanwhile, the IgM responses were short-term, and most patients sero-reverted within 2.5 months following symptom onset [8]. Patients with pneumonia, supplemental oxygen requirement, and intensive care unit admission had minimal neutralizing antibody decay at 180 days post symptoms onset (persistent group) compared to patients with the milder disease who showed seroreversion in less than 180 days. However, 27% of patients with pneumonia also sero-reverted in less than 180 days (rapid waning group) [43].

As the SARS-CoV-2 delta variant surge occurred in mid-2021, several hamlets and sub-districts in the study area implemented lockdown [44]. During the high transmission months, obstacles were encountered, which included patients' reluctance to enroll because of fear that

their infection status could be exposed. Patients also found no compelling reason to make a follow-up visit to the nearest healthcare facility after hospital discharge. Control group recruitment was also hindered by a reluctance to visit healthcare facilities, compounded by fears of getting infected. In addition, many potential control subjects refused testing because of fears that the test result would be positive. Recruitment of the control group is counted on the social and health system of the local community. As in many places in Indonesia, community leaders and health cadres have significant influence at the hamlet level. They would be the contact point because of their understanding and familiarity with their local community. By utilizing cadres or community leaders as the key persons to reach the potential respondents, the response is expected to be higher [45].

Progression into Acute Respiratory Distress Syndrome (ARDS) and multi-organ dysfunction are not the only problems leading to morbidity and mortality of COVID19 [46–48]. Clinical outcomes also depend on the adequacy of the clinical management. The lack of available resources, equipment, and properly trained healthcare workers, in addition to the shortage of medicines and limited healthcare infrastructure, have contributed to adverse outcomes [49–51]. Treatment variability described in the current study is due to multiple changes in the national clinical guidelines or other factors affecting the availability of resources (e.g., drugs and other supporting treatments). The variability may be reflected in the patients' characteristics and correlated with clinical outcomes.

This study has some limitations. As a long-term longitudinal study, retaining participants for the entire length of the 2-year study duration can be burdensome. The willingness of respondents to continue participating may decrease over time [52]. For those who refuse to visit a healthcare facility for monitoring, household visits will be done. This is expected to reduce attrition among participants [52, 53]. Another attempt to maintain participation is by offering incentives that include free basic medical check-ups, such as a registered nurse's examination of body temperature and blood pressure, for every follow-up visit [53]. Even though household visits is effective, this would lead to increased logistic burdens [52].

This study anticipated an absence in neutralizing antibodies (Nabs) measurement, arguably a better surrogate of immune protection [22, 54]. However, previous studies showed that Nabs levels are directly proportional to the antibody levels of RBD-IgG [8].

A previous study analyzed pseudo-neutralizing antibodies targeting the SARS-CoV-2 S protein in infected persons collected between 0- and 2.5-months post-symptoms. Over the course of infection, all individuals tested developed detectable Nabs levels. Nab titers were correlated with the concentration of anti-RBD IgG (r = 0.87). Furthermore, similar to anti-RBD IgG responses, Nab titers plateaued and remained detectable at later time points [8].

We also admit problems with comparing the screening methods for the patients and the controls. Limitations in study expenditure forced us to employ a more affordable screening method for the controls. We use the patients' breath-print analyses (GeNose C19®) instead of the nucleic acid amplification test (NAAT). GeNose C19® utilizes breath-borne volatile organic compound (VOC) biomarkers coupled with machine learning to discriminate between COVID-19 and non-COVID-19 respiratory infections accurately [55, 56]. Recent publications on GeNose showed a sensitivity of 86%-94% and specificity of 88%-95% [55]. A systematic review and meta-analysis study showed that VOCs-based breath analysis had a cumulative sensitivity of 98.2% (97.5% CI: 93.1%–99.6%) and specificity of 74.3% (97.5% CI: 66.4%–80.9%) [57].

Previous studies revealed public health challenges surrounding the stigma of being infected with COVID-19 in Malawi and Nigeria [58, 59]. Empirical observation of local communities in Sleman found that swabbing for PCR and antigen testing might have enhanced the stigma, where understandably, breath analysis would be a more acceptable option.

The study presented here will provide current and local evidence, even though it may not be generalizable on a global scale. The study outcomes will provide data for a better understanding of the serological measurement of SARS-CoV-2 infection, improving insights into the at-risk population and post-infection long-term follow-up for the kinetics of anti-SARS-CoV-2 antibody.

## Acknowledgments

We thank the Health and Demographic Surveillance System (HDSS) Sleman team for their administrative and field support and Prof. Nawi Ng for his valuable discussion.

## Author Contributions

**Conceptualization:** Eggi Arguni, Fatwa Sari Tetra Dewi, Jajah Fachiroh, Septi Kurnia Lestari, Teguh Haryo Sasongko, Lutfan Lazuardi.

**Data curation:** Eggi Arguni, Septi Kurnia Lestari.

**Formal analysis:** Eggi Arguni, Fatwa Sari Tetra Dewi, Jajah Fachiroh, Septi Kurnia Lestari, Lutfan Lazuardi.

**Investigation:** Dewi Kartikawati Paramita, Yogi Hasna Meisyarah, Merlinda Permata Sari, Zakiya Ammalia Farahdilla, Siswanto Siswanto.

**Methodology:** Eggi Arguni, Fatwa Sari Tetra Dewi, Jajah Fachiroh, Dewi Kartikawati Paramita, Septi Kurnia Lestari, Bayu Satria Wiratama, Lutfan Lazuardi.

**Project administration:** Annisa Ryan Susilaningrum, Bara Kharisma.

**Supervision:** Eggi Arguni, Bara Kharisma, Siswanto Siswanto.

**Writing – original draft:** Eggi Arguni, Fatwa Sari Tetra Dewi, Jajah Fachiroh, Septi Kurnia Lestari, Annisa Ryan Susilaningrum, Bara Kharisma, Yogi Hasna Meisyarah, Merlinda Permata Sari, Zakiya Ammalia Farahdilla, Lutfan Lazuardi.

**Writing – review & editing:** Eggi Arguni, Fatwa Sari Tetra Dewi, Jajah Fachiroh, Dewi Kartikawati Paramita, Septi Kurnia Lestari, Bayu Satria Wiratama, Annisa Ryan Susilaningrum, Bara Kharisma, Yogi Hasna Meisyarah, Merlinda Permata Sari, Zakiya Ammalia Farahdilla, Siswanto Siswanto, Muhammad Farhan Sjaugi, Teguh Haryo Sasongko, Lutfan Lazuardi.

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
