## [Decision Letter · Decision Letter 0]

8 Mar 2022

PONE-D-21-34739Two years antibody responses following SARS-CoV-2 infection in humans: a study protocolPLOS ONE

Dear Dr. Lazuardi,

Thank you for submitting your manuscript to PLOS ONE. After careful consideration, we feel that it has merit but does not fully meet PLOS ONE’s publication criteria as it currently stands. Therefore, we invite you to submit a revised version of the manuscript that addresses the points raised during the review process.

We look forward to receiving your revised manuscript.

Kind regards,

Harapan Harapan, MD, PhD

Academic Editor

PLOS ONE

Journal Requirements:

"This research is supported by the Indonesian Endowment Fund for Education and the Indonesian Science Fund through the International Collaboration RISPRO funding program (Research grant No. RISPRO/KI/B1/TKL/5/15867/2/2020)"

"The study has received financial grant support from the Ministry of Finance of the Republic of Indonesia LPDP (RISPRO/KI/B1/TKL/5/15867/2/2020).

EA received the award.

The funders had and will not have a role in study design, data collection and analysis, decision to publish, or preparation of the manuscript."

Reviewers' comments:

Reviewer's Responses to Questions

**Comments to the Author**

1. Does the manuscript provide a valid rationale for the proposed study, with clearly identified and justified research questions?

Reviewer #1: Yes

Reviewer #2: Yes

2. Is the protocol technically sound and planned in a manner that will lead to a meaningful outcome and allow testing the stated hypotheses?

Reviewer #1: Partly

Reviewer #2: No

3. Is the methodology feasible and described in sufficient detail to allow the work to be replicable?

Reviewer #1: No

Reviewer #2: Yes

4. Have the authors described where all data underlying the findings will be made available when the study is complete?

Reviewer #1: Yes

Reviewer #2: Yes

5. Is the manuscript presented in an intelligible fashion and written in standard English?

Reviewer #1: Yes

Reviewer #2: No

6. Review Comments to the Author

You may also provide optional suggestions and comments to authors that they might find helpful in planning their study.

Reviewer #1: Thank you for inviting me to review this manuscript. Although this manuscript seems interesting, there are several lacking points, especially within the Materials and Methods which need further clarification.

1) The number of references cited within the text paragraph should be ordered from their appearance in the text not from the alphabetical order. Please re-arrange all references within the text paragraph to match the journal's guidelines.

2) In Materials and Methods, I think between the exposed group and the control group is not comparable. If the participants in exposed group are those who have positive results from SARS-CoV-2 RT-PCR, then the control group should consist of those who have negative results from SARS-CoV-2 RT-PCR test. If the control group only use GeNose as diagnostic methods and not RT-PCR, it will cause bias in the results. As we all know, GeNose accuracy in diagnosing Covid-19 is still questionable, because it has low specificity and also may results in false negative results. Many of those who have negative GeNose results can still showed a positive results when tested with RT-PCR test as gold standard diagnostic methods. Therefore, the authors should revise this.

3) Moreover in the Materials and Methods, the inclusion criteria for the exposed groups is all patients with confirmed Covid-19 through RT-PCR test which can include all variety of age (pediatrics, teenagers, adults, older people). Meanwhile, in the control groups, the authors only include patients who are >18 years old. This can also cause bias in the results because of age gap between exposed and control groups. Please revise this.

4) In Materials and Methods, the authors should state about the guidelines they used in this study (e.g., STROBE guidelines).

5) In Materials and Methods, the technique used for calculation of the sample size should be explained more. The authors may look into these studies for references (1. http://dx.doi.org/10.1136/bmjopen-2021-050824, 2. https://doi.org/10.1186/s12889-018-5364-2)

6) In Materials and Methods (sample size), the authors have mentioned "To achieve the case-control ratios of 1 : 3, a correction for non-response, serologically positive control, and control who had a flu-like syndrome was made." What do the authors mean by this statement? Please explain more about this statement.

7) It would be better if the protocol of this study has been registered in international registries, such as ClinicalTrials.gov.

Reviewer #2: Manuscript Number: PONE-D-21-34739

Title: Two years antibody responses following SARS-CoV-2 infection in humans: a study protocol

Journal: Plos One

The author appraised this paper by observing the kinetics of anti-SARS134 CoV-2 antibody for 2 years after infection in relation to disease severity.

However, your article is inadequately presented. Furthermore, there are many grammatical mistakes and spelling mistakes as well. Although the article has scientific rigor, several major flows need to be corrected before publication.

Major comments:

General Comments:

1. Many non-scientific and incorrect/wrong information/sentences are there, which may mislead the readers.

2. Every section of the manuscript must be written scientifically according to the published literature with appropriate references.

3. English is weak. The authors need to improve their writing style. The whole manuscript needs to be checked by native English speakers (certificate mandatory).

3. Spacing, punctuation marks, grammar, and spelling errors should be reviewed wholly.

4. The research questions could be defined more precisely. This is probably due to language concerns.

Title:

1. The present title of the article looks okay.

Abstract:

1. The purpose and significance of this research must be explained in the abstract more clearly.

2. The abstract section is unsuitable—no focus point in the abstract section.

Introduction:

1. The introduction section is inapplicable. Need to change the introduction considerably. Try to include the existing research limitations also, how the present research unravels those limits.

2. The study's gaps should be clearly defined in the introduction section with the applicable references.

3. The flow of the introduction is not perfect and unspecific. My advice is to make the sentences more lucid and legible for more productive comprehension.

4. In the introduction section, there are many redundant sentences repeating in the whole paper, making me feel the paper like a "cut and paste" from several other resources.

5. The paragraphs are not logically arranged; there are unnecessary repeats.

6. Arrange the sentences: “There is an urgent need for assays that…”.

7. Many research programs still use paper-based…Not clear.

8. Delete I, we, our throughout the manuscript.

9. The first four paragraphs of the introduction section is not good. Need to merge and concise the whole introduction into four paragraphs.

10. The introduction section seems missing important information. This section needs profound modification with more Up-to-date references.

11. Aim of the study not clear.

12. The information which authors provided is old.

Materials and Methods:

1. Exclusion criteria…what was the purposes?

2. Blood samples collection and storage. Need details.

3. Data management plan: The writing is not good.

4. Safety considerations. Is this necessary?

5. Very few references are in the material and methods.

6. The type of data and statistical analyses planned: Not clear.

7. Patient and public involvement and ethics approval. Number is missing.

8. Need to add references for different sections.

Discussion:

1. The current discussion is poor.

2. unknown of sero-conversion and sero-reversion of antibodies: Need to elaborate.

3. Many text repetitions are found in the discussion section. It is highly recommended to emphasize findings and assumptions that support or disagree with other work(s).

4. Addendum, repetition of the results should be avoided in the discussion section.

5. It is still unclear how long the immunity lasts after patients recover from SARS-CoV2 infection…Is this okay?

6. Understanding about long-term antibody responses is currently limited…What’s the basis?

7. There had also been frequent adaptations…need more insights with references.

8. Challenges include a nationwide restriction: poor writings.

9. Our study targets a large sampling…. not justified.

Figures:

1. The resolution of figures can be improved.

References:

1. For better understanding, I feel this manuscript needs more detailed background information and a precise explanation of their study in the Introduction and Discussion section.

2. Several published articles related to antibody responses following SARS-CoV-2 infection in humans must be included within the relevant text part of the manuscript.

7. PLOS authors have the option to publish the peer review history of their article (what does this mean?). If published, this will include your full peer review and any attached files.

Reviewer #1: No

Reviewer #2: **Yes: **Talha Bin Emran

---

## [Author Response · Author response to Decision Letter 0]

19 Apr 2022

Thank you for giving us the opportunity to submit a revised version of our manuscript entitled, “Two years antibody responses following SARS-CoV-2 infection in humans: a study protocol” to PLOS ONE. 

We appreciate the time and effort that you and the reviewers have dedicated to providing your valuable feedback on our manuscript. I am grateful to the reviewers for their insightful comments on our paper. We have been able to incorporate changes to reflect most of the suggestions provided by the reviewers. We have highlighted the changes within the manuscript and you can identify the changes from track changes. Here is a point-by-point response to the editor and to the reviewers’ comments and concerns.

Comments from editor

Comment 1: Please ensure that your manuscript meets PLOS ONE's style requirements, including those for file naming. The PLOS ONE style templates can be found at https://journals.plos.org/plosone/s/file?id=wjVg/PLOSOne_formatting_sample_main_body.pdf and https://journals.plos.org/plosone/s/file?id=ba62/PLOSOne_formatting_sample_title_authors_affiliations.pdf

Response: We have revised the whole manuscript based on the guidelines given. Thank you for pointing this out. The changes can be found throughout the manuscript, including affiliations, font, paragraphing, spacing, figures, and references. We corrected the reference format and completed them by adding DOI. 

Comment 2: Please provide additional details regarding participant consent. In the ethics statement in the Methods and online submission information, please ensure that you have specified what type you obtained (for instance, written or verbal, and if verbal, how it was documented and witnessed). If your study included minors, state whether you obtained consent from parents or guardians. If the need for consent was waived by the ethics committee, please include this information

Response: Thank you for your important concern. We have, accordingly, specified the consent (line 268), with highlights. This study did not include minors.

Comment 3: We note that you have provided funding information that is not currently declared in your Funding Statement. However, funding information should not appear in the Acknowledgments section or other areas of your manuscript. We will only publish funding information present in the Funding Statement section of the online submission form. 

"The study has received financial grant support from the Ministry of Finance of the Republic of Indonesia LPDP (RISPRO/KI/B1/TKL/5/15867/2/2020).

EA received the award.

The funders had and will not have a role in study design, data collection and analysis, decision to publish, or preparation of the manuscript."

Response: Thank you for your suggestion. We deleted funding statement from the Acknowledgments section.

Comments from reviewer(s):

Reviewer #1

Comment 1: The number of references cited within the text paragraph should be ordered from their appearance in the text not from the alphabetical order. Please re-arrange all references within the text paragraph to match the journal's guidelines.

Response: Thank you for your valuable feedback. We edited the reference citation system according to the journal’s guideline.

Comment 2: In Materials and Methods, I think between the exposed group and the control group is not comparable. If the participants in exposed group are those who have positive results from SARS-CoV-2 RT-PCR, then the control group should consist of those who have negative results from SARS-CoV-2 RT-PCR test. If the control group only use GeNose as diagnostic methods and not RT-PCR, it will cause bias in the results. As we all know, GeNose accuracy in diagnosing Covid-19 is still questionable, because it has low specificity and also may result in false negative results. Many of those who have negative GeNose results can still showed a positive results when tested with RT-PCR test as gold standard diagnostic methods. Therefore, the authors should revise this.

Response: Thank you so much for your review. We put deep thought into this issue. 

We realize that the study design was not the highest standard. Ideally, both groups, case and control used the same diagnostic tool to screen SARS-CoV-2 infection. However, there were several aspects that we considered which are as follows:

1. When we developed the proposal, GeNose was set by our government as a screening tool for domestic travel requirements (such as trains, ships, planes) according to Formal Information Letter of Indonesia COVID-19 National Taskforce no 12 in April 2021, so it has been considered a commonly used screening tool in public and has higher acceptability than PCR or antigen test (by using respiratory swabs specimen).

2. Respiratory swab sampling has a negative stigma in our society (we also conveyed this in the discussion line 355-357). We have conducted a preliminary study, inviting healthy residents to be screened using a respiratory swab. However, out of the 15 residents who were invited, only 1 accepted the invitation. We repeated this method for groups of residents in different sub-districts, and the results were similar. We confirmed back to the candidates, that actually they did not refuse to participate in the research, but they refused to come because they did not want to be swab tested. 

3. In seeking control of healthy people, we invited candidate respondents who met the inclusion criteria to the public health center (Puskesmas) or a public hall close to their residence. Not all of these public places can be easily set-up to conduct a respiratory swab (it requires additional PPE, and the set-up process is inflexible). The stakeholders in charge of the public health center and public halls also suggested that we use other less invasive diagnostic tool to increase the recruitment rate, namely GeNose.

4. PCR or antibodies testing are much more costly than GeNose. For patients (case group), the testing cost is included in the regional outbreak health scheme. Nevertheless, by utilizing healthy people as controls, this cost must be borne by us, which is one of our limitations. As an illustration, at that time, the PCR test cost was around USD 95, the antigen swab was around USD 45, and GeNose was only USD 5.5. GeNose also showed a sensitivity of 86%-94% and specificity of 88%-95%. (Nurputra et al., 2021). We also conveyed this in the discussion section line 344-353.

5. Furthermore, a systematic review and meta-analysis study showed that VOCs-based breath analysis had a cumulative sensitivity of 98.2% (97.5% CI: 93.1%−99.6%) and specificity of 74.3% (97.5% CI: 66.4%−80.9%) (Subali et al., 2022).

6. We discussed the use of this GeNose as one of the weaknesses of the study.

Comment 3: Moreover in the Materials and Methods, the inclusion criteria for the exposed groups is all patients with confirmed Covid-19 through RT-PCR test which can include all variety of ages (pediatrics, teenagers, adults, older people). Meanwhile, in the control groups, the authors only include patients who are >18 years old. This can also cause bias in the results because of age gap between exposed and control groups. Please revise this.

Response: Thank you so much for your review. We revised inclusion criteria for case group (lines 148-151, highlighted):

“The inclusion criteria are patient age more than 18 years old who is confirmed COVID-19, defined as positive SARS-CoV-2 reverse transcriptase- polymerase chain reaction (RT-PCR) results from any respiratory sample, and reside in Sleman District at the time of diagnosis”

Comment 4: In Materials and Methods, the authors should state about the guidelines they used in this study (e.g., STROBE guidelines)

Response: Thank you for your suggestion. This study will be reported in accordance with the Strengthening the Reporting Observational Studies in Epidemiology (STROBE) checklist (lines 260-262, highlighted). 

Comment 5: In Materials and Methods, the technique used for calculation of the sample size should be explained more.

Response: Thank you for your review. We explained the calculation of the sample size (lines 139-144, highlighted).

“Fleiss’s formula was used to calculate the sample sizes of cases and controls with a ratio of 1:3 with 90% power and 95% confidence interval (CI) to detect an odds ratio of 2.30 The resulted sample size was 118 for cases. The sample size was further adjusted to account for the expected rate of lost to follow-up and missing data of 40%. Thus, the final sample size for cases is 165, and 495 healthy individuals for controls will be recruited from the Sleman population.”

Comment 6: In Materials and Methods (sample size), the authors have mentioned "To achieve the case-control ratios of 1 : 3, a correction for non-response, serologically positive control, and control who had a flu-like syndrome was made." What do the authors mean by this statement? Please explain more about this statement.

Response: Thank you so much for your review. We revised the whole paragraph (line 139-144, highlighted). In brief, we recruited 3 controls for each 1 case, with inclusion criteria they are healthy adults (aged >18 years old) with no prior or current SARS-CoV-2 infection upon screening (at the time of interview).

Comment 7: It would be better if the protocol of this study has been registered in international registries, such as ClinicalTrials.gov

Response: Thank you for your suggestion. We are in the process register to ClinicalTrials.gov through the administrator at our university.

Reviewer #2

General Comments:

Comment 1: Many non-scientific and incorrect/wrong information/sentences are there, which may mislead the readers.

Response: Thank you. We revised the whole manuscript, and consulted this manuscript to native English speaker registered in our university.

Comment 2: Every section of the manuscript must be written scientifically according to the published literature with appropriate references.

Response: Thank you. We added recent references to support introduction/background and discussion sections

Comment 3: English is weak. The authors need to improve their writing style. The whole manuscript needs to be checked by native English speakers (certificate mandatory). Spacing, punctuation marks, grammar, and spelling errors should be reviewed wholly.

Response: Thank you. We consulted this manuscript to native English speaker registered in our university. The certificate is attached to this letter.

Comment 4: The research questions could be defined more precisely. This is probably due to language concerns.

Response: Thank you for your suggestion. We incorporate the research question together with the aim of the study which can be found in the abstract (line 39-42, highlighted).

“We aim to prospectively observe and determine the kinetics of the anti SARS-CoV-2 antibody for 2 years after infection in relation to disease severity and to determine the risk and protective factors of SARS CoV-2 infections in the community.”

and in the ‘Aims of the study’ section (line 98-107).

“The primary objective of this study is to prospectively observe and determine the kinetics of anti-SARS-CoV-2 antibody for 2 years after infection in relation to disease severity. The secondary objective is to identify the risk and protective factors of SARS CoV-2 infections through comparison of SARS-CoV-2 confirmed patients (case group) and their respective neighborhood individuals (control group).”

Title:

Comment 1: The present title of the article looks okay.

Response: Thank you for your positive review

Abstract:

Comment 1: The purpose and significance of this research must be explained in the abstract more clearly.

Response: Thank you. We revised the abstract. The purposes are stated in line 39-42 (highlighted) and the significance of current study was stated in line 36-39 (highlighted).

“The long-term antibody response to the novel SARS-CoV-2 in infected patients and their residential neighborhood remains unknown In Indonesia. This information will provide insights into the antibody kinetics over a relatively long period as well as transmission risk factors in the community. We aim to prospectively observe and determine the kinetics of the anti SARS-CoV-2 antibody for 2 years after infection in relation to disease severity and to determine the risk and protective factors of SARS CoV-2 infections in the community.”

Comment 2: The abstract section is unsuitable—no focus point in the abstract section.

Response: Thank you for your correction. We edited the abstract structure according to journal’s guidelines.

Introduction 

Comment 1: The introduction section is inapplicable. Need to change the introduction considerably. Try to include the existing research limitations also, how the present research unravels those limits.

Response: Thank you for your suggestions. As you see in our revision, we re-structured the introduction majorly. Other studies observed not more than 13 months, while the current study will monitor immune response until two years following infection.

Comment 2: The study's gaps should be clearly defined in the introduction section with the applicable references.

Response: the biggest gap of this field is that seroconversion and duration of immune response among patients with COVID-19 in Indonesia, particularly in a cohort setting, has never been reported (line 85-86, highlighted)

“Seroconversion and duration of immune response among patients with COVID-19 in Indonesia, particularly in a cohort setting, has never been reported.”

Comment 3: The flow of the introduction is not perfect and unspecific. My advice is to make the sentences more lucid and legible for more productive comprehension.

Response: Thank you for your constructive comment. We revised the introduction section majorly.

Comment 4: In the introduction section, there are many redundant sentences repeating in the whole paper, making me feel the paper like a "cut and paste" from several other resources.

Response: Thank you. We re-structured the introduction and made the better flow of the story as a background of the study

Comment 5: The paragraphs are not logically arranged; there are unnecessary repeats.

Response: Thank you, we deleted any repetition and re-arranged the whole introduction.

Comment 6: Arrange the sentences: “There is an urgent need for assays that…”.

Response: We deleted that sentence and replaced it with a new paragraph (line 211-223, highlighted, tracked changes)

Comment 7: Many research programs still use paper-based…Not clear.

Response: We deleted that sentence and replaced it with a new Introduction section (line 56-93).

Comment 8: Delete I, we, our throughout the manuscript.

Response: Thank you for your suggestion. We deleted I, we and our throughout the manuscript.

Comment 9: The first four paragraphs of the introduction section is not good. Need to merge and concise the whole introduction into four paragraphs.

Response: Thank you for your suggestion. We revised the Introduction section thoroughly (line 56-93).

Comment 10: The introduction section seems missing important information. This section needs profound modification with more Up-to-date references.

Response: We revised the introduction thoroughly, and we added up-to-date references. (line 152-231)

Comment 11: Aim of the study not clear.

Response: Thank you so much for your input. Please refer to our response to your general comments, comment#4.

Comment 12: The information which authors provided is old.

Response: Thank you for your comment. We deleted several references and replaced them with recent references. 

Materials and Methods:

Comment 1: Exclusion criteria…what was the purposes? 

Response: Thank you for the question. We revised the exclusion criteria (line 155-157), Those who have had immunosuppressive conditions, or immunodeficiency disease, or is currently receiving any immunosuppressive therapy were not our exclusion criteria.

“The exclusion criteria are any eligible person who shows a flu-like syndrome on the screening day.”

Comment 2: Blood samples collection and storage. Need details.

Response: Thank you for your suggestion. We revised the blood samples collection and storage section (line 197-207, highlighted).

Comment 3: Data management plan: The writing is not good.

Response: Thank you for your comment. We revised data management plan section (line 227-242, highlighted).

“Data management plan

 All respondents will have unique identification. Research nurse/field officers will enter data into eSynthesis (version 1.9.8 - A survey tool. Sleman, Yogyakarta, Indonesia), an application, that have been installed on their mobile gadget tablet. Supervisors and data managers will check the suitability of the data to ensure data quality. Maintaining data quality will include calibrating instruments regularly, and having a validation feature in the application. A red warning sign will appear if the input format does not match or if there are data that have not been filled in. The data will be cleaned several times by the data manager until ready for analysis. Data types consist of text, number, audio, and image data. The data consist of primary and secondary data. Primary data will be obtained by measuring directly and interviewing respondents in the field. Secondary data will be taken from the respondents' medical records. Data will be stored on Google Drive and institutional servers which are restricted to a particular research team. Data processing (collection, storage, use) in the study is based on the UGM Academic Hospital and the Health and Demographic Surveillance System Sleman (HDSS Sleman) data protection concepts.”

Comment 4: Safety considerations. Is this necessary? 

Response: Thank you for pointing this out. We agree with your comment, and we deleted that sub section of safety consideration, accordingly.

Comment 5: Very few references are in the material and methods.

Response: Thank you. We revised the Material and methods section majorly, and added supportive references. 

Comment 6: The type of data and statistical analyses planned: Not clear. 

Response: Thank you. We revised the type of data and statistical analyses planned section. Please refer to line 252-263 (highlighted).

“The type of data and statistical analyses planned 

 Continuous variables with normal distribution will be reported as mean with standard deviation (SD), and non-normal distributed variables as median with interquartile range (IQR). The Kaplan-Meier method will be used to calculate cumulative seroconversion rates. The association between antibody level and disease severity will be estimated by logistic regression. The characteristics of the case and control groups will be described as mean and SD or frequency and percentage. The factors (risk and protective) associated with the COVID-19 infection will be determined using conditional (matched) logistic regression and presented as OR and its 95% CI. This study will be reported in accordance with the Strengthening the Reporting Observational Studies in Epidemiology (STROBE) checklist.36 Missing data and lost to follow-up patients will be reported. Missing data will be handled by using multiple imputation models.37 “

Comment 7: Patient and public involvement and ethics approval. Number is missing.

Response: Than you, the number of ethics approval was mentioned in line 266-269 (highlighted).

“This study was reviewed and approved by the Medical and Health Research Ethics Committee (MHREC) Faculty of Medicine, Public Health, and Nursing UGM (KE/FK/0882/EC/2020). Written informed consent will be from all respondents prior to enrolment into the study.”

Comment 8: Need to add references for different sections.

Response: Thank you for your suggestion. We added supporting references in every sub-section in the Methods.

Discussion:

Comment 1: The current discussion is poor.

Response: Thank you for your valuable comment. We revised the Discussion section majorly as you can refer to the line 279-362.

Comment 2: unknown of sero-conversion and sero-reversion of antibodies: Need to elaborate.

Response: Thank you. We elaborated this in the line 288-297 (highlighted).

“Potential key points relevant to the current pandemic include unknown seroconversion and seroreversion in hospitalized patients. Time to seroconversion may be related to the severity of disease. A study from North America showed that in symptomatic COVID-19 patients, the median time to seroconversion of SARS-CoV-2 RBD (IgG and IgM) among hospitalized patients was 4 days earlier compared to non-hospitalized patients, proposing an association between antibody kinetics and disease severity.8 IgG responses to the SARS-CoV-2 spike RBD showed minimal seroreversion, which were sustained for more than 3 months. Meanwhile, the IgM responses were short-term, and most patients seroreverted within 2.5 months following symptom onset.8”

Comment 3: Many text repetitions are found in the discussion section. It is highly recommended to emphasize findings and assumptions that support or disagree with other work(s).

Response: Thank you so much for your input. We revised the Discussion section majorly, and we added previous reports those support our study, since our current study is still on going.

Comment 4: Addendum, repetition of the results should be avoided in the discussion section.

Response: Thank you for your valuable input. We revised and deleted any repetition. Our study is still on going, so we still do not report any result.

Comment 5: It is still unclear how long the immunity lasts after patients recover from SARS-CoV2 infection…Is this okay?

Response: Thank you for your comment. We deleted that sentence and moved to that idea in introduction section (line 57-86).

Comment 6: Understanding about long-term antibody responses is currently limited…What’s the basis? 

Response: Thank you. We deleted that sentence and moved that idea into the introduction section (line 73-86, highlighted), especially long-term antibody response in correlation with severity. 

Comment 7: There had also been frequent adaptations…need more insights with references.

Response: Thank you for your suggestions. We added new references to support our discussion.

Comment 8: Challenges include a nationwide restriction: poor writings.

Response: Thank you for your comment. We revised the paragraph as you can find in line 299-302 (highlighted).

“As the SARS-CoV-2 delta variant surge occurred in mid-2021, several hamlets and sub-districts in the study area implemented lockdown.39 During the high transmission months, obstacles were encountered, that included patients’ reluctance to enroll because of fear that their infection status could be exposed”

Comment 9: Our study targets a large sampling…. not justified. 

Response: Thank you, we revised the whole paragraph (line 358-362, highlighted).

“The study presented here will provide current and local evidence, even though it may not be generalizable on a global scale. The study outcome will provide data for better understanding in serological measurement of SARS-CoV-2 infection which will improve insights into at-risk population and post-infection long-term follow-up for the kinetics of anti-SARS-CoV-2 antibody.”

Figures: 

Comment 1: The resolution of figures can be improved.

Response: Thank you for your suggestion. We uploaded the figure files to the Preflight Analysis and Conversion Engine (PACE) digital diagnostic tool to ensure that figure meets PLOS requirements.

References:

Comment 1: For better understanding, I feel this manuscript needs more detailed background information and a precise explanation of their study in the Introduction and Discussion section.

Response: Thank you for your valuable suggestion. We agree with your comment, therefore we revised the introduction and discussion section majorly.

Comment 2: Several published articles related to antibody responses following SARS-CoV-2 infection in humans must be included within the relevant text part of the manuscript.

Response: Several supporting and up-to-date references already added to the manuscript. Those references are as follow:

Crawford KH, Dingens AS, Eguia R, Wolf CR, Wilcox N, Logue JK, et al. Dynamics of neutralizing antibody titers in the months after severe acute respiratory syndrome Coronavirus 2 infection. J Infect Dis. 2021;223(2):197-205. doi: 10.1093/infdis/jiaa618

Wang H, Yuan Y, Xiao M, Chen L, Zhao Y, Zhang H, et al. Dynamics of the SARS-CoV-2 antibody response up to 10 months after infection. Cell Mol Immunol. 2021;18(7):1832-1834. doi: 10.1038/s41423-021-00708-6

Wang K, Long QX, Deng HJ, Hu J, Gao QZ, Zhang GJ. et al. Longitudinal dynamics of the neutralizing antibody response to severe acute respiratory syndrome coronavirus 2 (SARS-CoV-2) infection. Clin Infecti Dis. 2021;73(3):e531-9. doi: 10.1093/cid/ciaa1143

Liu C, Yu X, Gao C, Zhang L, Zhai H, Hu Y, et al. Characterization of antibody responses to SARS‐CoV‐2 in convalescent COVID‐19 patients. J Med Virol. 2021;93(4):2227-2233. doi: 10.1002/jmv.26646

Feng C, Shi J, Fan Q, Wang Y, Huang H, Chen F, et al. Protective humoral and cellular immune responses to SARS-CoV-2 persist up to 1 year after recovery. Nat Commun. 2021;12(1):1-7. doi: 10.1038/s41467-021-25312-0

Gallais F, Gantner P, Bruel T, Velay A, Planas D, Wendling MJ, et al. Evolution of antibody responses up to 13 months after SARS-CoV-2 infection and risk of reinfection. EBioMedicine. 2021;71:103561. doi: 10.1016/j.ebiom.2021.103561

Lumley SF, Wei J, O’Donnell D, Stoesser NE, Matthews PC, Howarth A, et al. The duration, dynamics, and determinants of Severe Acute Respiratory Syndrome Coronavirus 2 (SARS-CoV-2) antibody responses in individual healthcare workers. Clin Infect Dis. 2021;73(3):e699-709. doi: 10.1093/cid/ciab004

Petersen MS, Hansen CB, Kristiansen MF, Fjallsbak JP, Larsen S, Hansen JL, et al. SARS-CoV-2 natural antibody response persists for at least 12 months in a nationwide study from the Faroe Islands. Open Forum Infect Dis. 2021;8(8):ofab378. doi: 10.1093/ofid/ofab378

Varona JF, Madurga R, Peñalver F, Abarca E, Almirall C, Cruz M, et al. Kinetics of anti-SARS-CoV-2 antibodies over time: results of 10 month follow up in over 300 seropositive health care workers. Eur J Intern Med. 2021;89:97-103. doi: 10.1016/j.ejim.2021.05.028

Dan JM, Mateus J, Kato Y, Hastie KM, Yu ED, Faliti CE, et al. Immunological memory to SARS-CoV-2 assessed for up to 8 months after infection. Science. 2021;371(6529):eabf4063. doi: 10.1126/science.abf4063

Anand SP, Prévost J, Nayrac M, Beaudoin-Bussières G, Benlarbi M, Gasser R, et al. Longitudinal analysis of humoral immunity against SARS-CoV-2 spike in convalescent individuals up to 8 months post-symptom onset. Cell Rep Med. 2021;2(6):100290. doi: 10.1016/j.xcrm.2021.100290

In addition to the above comments, all spelling and grammatical errors pointed out by the reviewers have been corrected. 

We look forward to hearing from you in due time regarding our submission and to respond to any further questions and comments you may have. 

Sincerely, 

Lutfan Lazuardi

---

## [Decision Letter · Decision Letter 1]

8 Jul 2022

PONE-D-21-34739R1Two years antibody responses following SARS-CoV-2 infection in humans: a study protocolPLOS ONE

Dear Dr. Lazuardi,

Thank you for submitting your manuscript to PLOS ONE. After careful consideration, we feel that it has merit but does not fully meet PLOS ONE’s publication criteria as it currently stands. Therefore, we invite you to submit a revised version of the manuscript that addresses the points raised during the review. 

We look forward to receiving your revised manuscript.

Kind regards,

Harapan Harapan, MD, PhD

Academic Editor

PLOS ONE

Journal Requirements:

Reviewers' comments:

Reviewer's Responses to Questions

**Comments to the Author**

1. Does the manuscript provide a valid rationale for the proposed study, with clearly identified and justified research questions?

Reviewer #1: Yes

Reviewer #2: Yes

2. Is the protocol technically sound and planned in a manner that will lead to a meaningful outcome and allow testing the stated hypotheses?

Reviewer #1: Yes

Reviewer #2: Yes

3. Is the methodology feasible and described in sufficient detail to allow the work to be replicable?

Reviewer #1: Yes

Reviewer #2: Yes

4. Have the authors described where all data underlying the findings will be made available when the study is complete?

Reviewer #1: Yes

Reviewer #2: Yes

5. Is the manuscript presented in an intelligible fashion and written in standard English?

Reviewer #1: Yes

Reviewer #2: Yes

6. Review Comments to the Author

You may also provide optional suggestions and comments to authors that they might find helpful in planning their study.

Reviewer #1: Thank you for revising the manuscript according to my comments. The revision has made significant improvement into the overall quality of the manuscript, especially in the Methodology section. I think the manuscript can now be accepted for publication.

Reviewer #2: Manuscript Number: PONE-D-21-34739R1

Title: Two years antibody responses following SARS-CoV-2 infection in humans: a study protocol

Journal: Plos One

The author appraised this paper by observing the kinetics of anti-SARS-CoV-2 antibody for 2 years after infection in relation to disease severity.

Although the article has scientific rigor, several minor flows need to be corrected before publication.

Minor comments:

Introduction:

1. The introduction section seems missing important information. This section needs profound modification with more Up-to-date references.

Discussion:

1. The discussion section can improve.

7. PLOS authors have the option to publish the peer review history of their article (what does this mean?). If published, this will include your full peer review and any attached files.

Reviewer #1: No

Reviewer #2: No

---

## [Author Response · Author response to Decision Letter 1]

25 Jul 2022

Dear Academic Editor

PLOS ONE

Thank you for giving us the opportunity to submit a second round revised draft of our manuscript titled “Two years antibody responses following SARS-CoV-2 infection in humans: a study protocol” to PLOS ONE. 

Again, we appreciate the time and effort that you and the reviewers have dedicated to providing your valuable feedback after our first response. We have highlighted the changes within the manuscript and you can identify the changes from track changes. Here is a point-by-point response to the reviewers’ comments and concerns.

Reviewer #1: Reviewer #1: Thank you for revising the manuscript according to my comments. The revision has made significant improvements to the overall quality of the manuscript, especially in the Methodology section. I think the manuscript can now be accepted for publication.

Response: Thank you so much for your positive comment.

Reviewer #2: Although the article has scientific rigor, several minor flows need to be corrected before publication.

Minor comments:

Comment 1:

Introduction: The introduction section seems missing important information. This section needs profound modification with more Up-to-date references.

Response: Thank you for your valuable feedback. We revised the introduction section (highlighted)

Comment 2:

Discussion: The discussion section can improve.

Response: Thank you so much. We appreciate your comment. We revised the discussion section (highlighted)

We look forward to hearing from you in due time regarding our re-submission.

Sincerely, 

Lutfan Lazuardi, MD, PhD

Corresponding author

Juli 16, 2022

---

## [Editor Report · Decision Letter 2]

26 Jul 2022

Two years antibody responses following SARS-CoV-2 infection in humans: a study protocol

PONE-D-21-34739R2

Dear Dr. Lazuardi,

We’re pleased to inform you that your manuscript has been judged scientifically suitable for publication and will be formally accepted for publication once it meets all outstanding technical requirements.

Kind regards,

Harapan Harapan, MD, PhD

Academic Editor

PLOS ONE
---

## [Editor Report · Acceptance letter]

4 Aug 2022

PONE-D-21-34739R2 

Two-years antibody responses following SARS-CoV-2 infection in humans: a study protocol 

Dear Dr. Lazuardi:

I'm pleased to inform you that your manuscript has been deemed suitable for publication in PLOS ONE. Congratulations! Your manuscript is now with our production department. 

Kind regards, 

on behalf of

Dr. Harapan Harapan 

Academic Editor

PLOS ONE